# Characterization of a *Candida albicans* Mutant Defective in All MAPKs Highlights the Major Role of Hog1 in the MAPK Signaling Network

**DOI:** 10.3390/jof6040230

**Published:** 2020-10-17

**Authors:** Inês Correia, Duncan Wilson, Bernhard Hube, Jesús Pla

**Affiliations:** 1iBiMED-Institute of Biomedicine, Department of Medical Sciences, University of Aveiro, Agra do Crasto, 3810-193 Aveiro, Portugal; 2Medical Research Council Centre for Medical Mycology, School of Biosciences, University of Exeter, Exeter EX4 4QD, UK; Duncan.Wilson@exeter.ac.uk; 3Department of Microbial Pathogenicity Mechanisms, Leibniz Institute for Natural Product Research and Infection Biology-Hans-Knoell-Institute, Beutenbergstraße 11A, 07745 Jena, Germany; bernhard.hube@hki-jena.de; 4Institute of Microbiology, Friedrich Schiller University, Neugasse 25, 07743 Jena, Germany; 5Departamento de Microbiología y Parasitología-IRYCIS, Facultad de Farmacia, Universidad Complutense de Madrid, Avda. Ramón y Cajal s/n, 28040 Madrid, Spain

**Keywords:** MAPK, osmotic stress, filamentation, chlamydospore, cell wall, oxidative stress, fungal virulence

## Abstract

The success of *Candida albicans* as a pathogen relies on its ability to adapt and proliferate in different environmental niches. Pathways regulated by mitogen-activated protein kinases (MAPKs) are involved in sensing environmental conditions and developing an accurate adaptive response. Given the frequent cooperative roles of these routes in cellular functions, we have generated mutants defective in all combinations of the four described MAPKs in *C. albicans* and characterized its phenotype regarding sensitiveness to specific drugs, morphogenesis and interaction with host immune cells. We demonstrate that all MAPKs are dispensable in this yeast as a mutant defective in Cek1, Cek2, Mkc1 and Hog1 is viable although highly sensitive to oxidative and osmotic stress, displaying a specific pattern of sensitivity to antifungals. By comparing its phenotype with single, double and triple combinations of MAPK-deletion mutants we were able to unveil a Cek1-independent mechanism for Hog1 resistance to Congo red, and confirm the predominant effect of Hog1 on oxidative and osmotic adaptation. The quadruple mutant produces filaments under non-inducing conditions, but is unable to develop chlamydospores. Furthermore, *cek1 cek2 mkc1 hog1* cells switch to the opaque state at high frequency, which is blocked by the ectopic expression of *HOG1* suggesting a role of this kinase for phenotypic switching.

## 1. Introduction

*Candida albicans* is a pathogenic fungus frequently causing infection in humans. However, 30 to 70% of the human population is colonized by this microorganism in the vaginal and gastrointestinal tract without any symptoms. Only when the host immune system is compromised, when the microbiota is disturbed or when the epithelial barrier integrity is damaged, *C. albicans* may proliferate inside or outside these niches and can, in severe cases, develop invasive infections that are associated with a high mortality rate [1,2,3].

An important feature of the biology of *C. albicans* is its ability to change its morphology from yeast-to-hypha in response to nutritional and environmental factors, such as temperature (37 °C), pH (>7), or the presence of N-acetylglucosamine or serum. The filamentous form has been considered for a long time as the more invasive morphology of the fungus as mutants unable to filament under *in vitro* conditions (such as *efg1* and *cph1*) showed impaired virulence in animal models [4]. However, both morphologies are important for infection: yeast cells for fungal dissemination and hyphae for tissue invasion, host cell damage and immune evasion [5,6,7,8].

Under specific laboratory conditions such as absence of light, glucose limitation, low temperature and microaerophilia [9,10], *C. albicans* can also form thick-walled structures called chlamydospores. Although their biological function is unknown, they have been proposed to play a role in persistence (survival under adverse conditions) rather than dissemination [11,12,13]. Different genes were shown to be required for chlamydospore formation, including *EFG1* and *HOG1* [14,15], and specific markers have been identified [16].

For decades, *C. albicans* was thought to be an asexual yeast. However, under certain circumstances, mating is possible [17,18,19]. This requires conversion of the mating-type locus (MTL a/α) to a/a or α/α and an epigenetic switch called white-opaque switching, a spontaneous and reversible change from the ubiquitous white type of cells to the opaque phenotype. This triggers a pheromone-mediated process that can ultimately result in cell fusion [17,18,19]. White-opaque switching has been the subject of intensive study in recent years [20] as mating is a source of genetic variability with potential implications on cells adaptation to new environments [21,22,23]. Notably, gene flow has recently been detected between *C. albicans* isolates [24].

All these morphological transitions, as well as the adaptation to environmental stress, are tightly controlled by different signaling pathways. In *C. albicans*, four mitogen-activated protein kinases (MAPKs) mediate signal transduction. Hog1 regulates adaptation to oxidative and osmotic stress [15,25,26], Mkc1 is involved in cell wall construction (cell integrity pathway) [27] and invasion of solid surfaces [28], Cek1 is implicated in cell wall biogenesis [29,30,31], while Cek2 is involved in mating [32]. However, the interconnections among these pathways are frequent and increase the plasticity and complexity of the response [33,34,35]. For example, Mkc1, like Hog1, is also phosphorylated in response to oxidative stress [36]. Cek1 and Cek2 are both important for mating and cell wall damage response [31,37]. Cek1 and Hog1 MAPKs play complementary roles on cell wall biogenesis and in chlamydospore formation: the resistance of a *hog1* mutant to Congo red, calcofluor white and nikkomycin Z [25] seems to be caused by Cek1 hyperactivation and the inability of *hog1* to form chlamydospores is suppressed by the additional deletion of upstream elements of the Cek1 pathway such as *CST20* or *HST7* [38]. All these examples illustrate the intricacy and crosstalk among these routes and the cooperative and counterbalancing role that they have in the cell.

To further elucidate the crosstalk between all these central MAPK signaling networks, we have generated a strain deprived of the four main MAPKs. Here, we characterized this strain and assessed the importance of each MAPK in specific stress conditions as well as in the morphogenetic programs of *C. albicans*.

## 2. Materials and Methods

### 2.1. Strain Construction and Growth Conditions

*C. albicans* strains used in this work are listed in Table 1. All strains from the MAPK mutants collection were derived from the laboratory-adapted CAI4 and were obtained using the *URA3*-blaster protocol [39]. *HOG1*, *CEK2* and *MKC1* deletions were made as previously described [26,27,38] and correct disruptions were confirmed by Southern blot (Appendix A). MAPK’s ectopic expression was achieved by integration at the *ADH1* locus of the plasmid pNRU [37]. *CEK1*, *CEK2*, *MKC1* and *HOG1* ORFs were amplified by PCR from the clinical isolate SC5314. The correct expression of these epitope-labelled alleles was verified by Western blots against the myc epitope. The CAI4 parental strain CAF2, which is heterozygous for *URA3*, has been used as the background wild type.

Yeast strains were routinely grown at 37 °C in yeast extract–peptone–dextrose (YPD) medium (2% glucose, 2% peptone, 1% yeast extract) and short term stored at 4 °C. Analysis of strains’ sensitivity to different agents was performed through drop tests, by spotting 5 µL of tenfold serial dilutions of overnight grown cultures on solid YPD plates supplemented (or not, control) with the indicated compound at the indicated concentration. Plates were incubated for 48 h at 37 °C (unless otherwise stated) and scanned for Figure assembly. For analysis of yeast sensitivity to antifungals, cells from overnight cultures grown at 37 °C were inoculated in 96-well plates at O.D. (Optical Density) = 0.025 in medium supplemented with increased concentrations of the compound and incubated statically for 24 h at 37 °C. The final O.D. of each strain in the presence of different compound concentrations was measured and normalized to the final O.D. reached by the positive control (strains inoculated in medium without the antifungal).

### 2.2. Interaction Assays with Murine Macrophages

*C. albicans* killing by murine macrophages from cell line RAW 264.7 was performed in 24-well plates. One million immune cells were co-incubated with yeast cells (1 yeast cell:20 phagocytes) for 4 h at 37 °C and 5% CO_2_. After phagocyte lysis with water and use of a scrapper to recover adhered cells, serial dilutions were made and spread over YPD agar plates for determinations of CFUs (Colony Forming Units). Plates were incubated for 24 h at 37 °C. The percentage of killing for each strain was expressed as the reduction in CFUs from yeast co-cultures with macrophages versus cultures containing only yeast cells without phagocytes made simultaneously. A one-way ANOVA multiple comparison test was used to assess significance related to CAF2.

### 2.3. Morphogenesis Assays

Hyphal formation in liquid medium was induced by serum 100% or 5%: 10^6^ cells from overnight grown cultures were inoculated in YPD medium supplemented or not with serum and grown at 30 °C. Cells were microscopically observed after 5 h and 24 h of growth. Pictures were taken and processed likewise. To induce chlamydospore formation, 25 to 50 cells from a stationary culture at 37 °C were inoculated in corn meal agar (CMA) supplemented with 1% Tween 80 and covered by coverslips to induce microaerophilia. The samples were incubated in the dark at 24 °C for up to 10 days. Chlamydospores were observed at day 4 and 7. White-opaque switching of *C. albicans* strains was induced by growing cells on solid media at room temperature (<21 °C) for at least three days. Strains previously grown at 30 °C were spread onto YPD plates supplemented with phloxine B (50 µg/mL) and around 400 cells were analyzed. White-opaque reversibility was studied by incubating opaque cells at different temperatures on solid medium (YPD-phloxine B).

## 3. Results

### 3.1. A Strain Lacking CEK1, CEK2, MKC1 and HOG1 Is Viable but Highly Sensitive to Osmotic and Oxidative Stress

Previous studies by our group and others have revealed the crucial role that efficient MAPK-mediated signaling plays in response to stress. In order to understand and uncover putative crosstalk between MAPKs, we expanded our previously described collection of single- and double-MAPK mutants [34,37] by deleting the remaining MAPKs in order to generate all combinations of triple mutants and a strain defective in all four described MAPKs in this fungus. This strain is viable and does not display evident growth defects under standard laboratory conditions. We analyzed this strain in the presence of specific compounds known to activate the pathways and compared its behavior with strains that have only one functional MAPK, either endogenous or ectopically integrated, using standard drop assays on solid media. The results are shown in Figure 1 and the main findings are the following. First, the quadruple mutant showed high sensitivity to both osmotic and oxidative stress. Second, none of the mutants with an intact HOG pathway were found to be sensitive to NaCl or Sorbitol (Figure 1A) reinforcing the idea that Hog1 is the key kinase for adaptation to osmotic stress; in fact, the sensitivity of the *cek1 cek2 mkc1 hog1* mutant to NaCl was completely rescued to wild-type levels by integration of a plasmid carrying *HOG1*. Third, the *cek1 cek2 mkc1 hog1* mutant strain showed the same phenotype as the one observed for all strains defective in both Cek1 and Hog1 kinases, which is a hypersensitivity to osmotic stress (Figure 1A, NaCl 1M) [37,44] and the additional deletion of other kinases did not alter the levels of growth inhibition. Accordingly, the additional deletion of *CEK1* aggravates the growth defects of a *hog1 mkc1 cek2* strain which is by itself sensitive, confirming that Cek1 plays a crucial role in *hog1* defective strains to survive under hyperosmotic conditions. Interestingly, all mutants from the collection lacking *HOG1* were found to be similarly sensitive to oxidants (Figure 1B), strongly supporting the idea that Hog1 is the main MAPK responsible for *C. albicans* survival under these conditions.

Interaction studies of kinase mutants with immune cells have revealed an increased susceptibility of *hog1* to murine macrophage killing (probably due to the formation of reactive oxygen species (ROS)), as well as a slight resistance of *cek1* with no altered susceptibility of *cek2* and *mkc1* compared to wild-type controls [43]. We therefore analyzed the behavior of our mutant collection when confronted with macrophages. For this, the mutants were individually incubated with a confluent monolayer of RAW 264.7 cell line macrophages infected at MOI 1:20 (yeast:macrophages) for 4 h.

We observed an increased susceptibility of all triple mutants compared to the strains defective in only two kinases and, in turn, an increased susceptibility for the quadruple mutant as compared to the triple deletion mutants (Figure 2). In fact, the absence of all MAPKs led to a 2.8-fold increase in *C. albicans* mutant cell susceptibility to this macrophage cell line. Interestingly, susceptibility to macrophages was not exclusive to *hog1* mutants as *cek1 cek2, cek1 mkc1* and particularly *cek1 cek2 mkc1* also showed an increased vulnerability to macrophage-mediated killing.

To extend our analysis on the mutants’ susceptibility to stress conditions, we assessed their growth inhibition in the presence of amphotericin B and fluconazole which directly or indirectly target the fungal cell membrane. Amphotericin B binds to ergosterol causing permeability alterations and eventually pore formation, while fluconazole inhibits the ergosterol biosynthesis leading to the accumulation of toxic sterol intermediates and cell membrane stress. Despite the fact that these antifungals act through different mechanisms of action and with different outcomes (amphotericin B is fungicidal while fluconazole is fungistatic) they both induce ROS production in *C. albicans* [45,46,47]. Our data support these results as all strains with a *HOG1* deletion were found to be sensitive to amphotericin B (Figure 3).

While the deletion of *MKC1* (in any background) did not affect the susceptibility to amphotericin B, strains defective in *CEK1* and/or *CEK2* showed a reduced susceptibility to the compound. The additional deletion of *HOG1* in mutants lacking *CEK1* and/or *CEK2*, however, impaired their growth. Therefore, the quadruple mutant was highly sensitive to amphotericin B, which was reverted by *HOG1* ectopic expression to the levels obtained for the wild type [48]. Hog1 also seems to be crucial for *C. albicans* adaptation to fluconazole. Amongst the single mutants, *hog1* showed the most significant growth defect and all other *hog1*-null strains displayed a clear sensitive phenotype (Figure 3). Altogether, these results indicate that *C. albicans* adaptation to osmotic, oxidative and antifungal drug conditions is highly dependent on the Hog1 kinase while other MAPKs play secondary roles for these conditions.

### 3.2. Hog1 Resistance to Cell Wall Interfering Drugs Is Not Exclusively Due to Cek1 Hyperactivation

Signaling pathways mediated by MAPKs have also a crucial role in the biogenesis and maintenance of the fungal cell wall. We therefore analyzed our deletion mutant collection for their behavior in the presence of Congo red. While the *mkc1* and *cek2* mutants have been reported to show sensitivity [27,37], the triple mutant *hog1 mkc1 cek2* showed a resistant phenotype (Figure 4A) therefore resembling the *hog1* single mutant [25]. This is further evidence that Hog1 is the dominant MAPK governing stress responses in *C. albicans*.

However, the additional deletion of *CEK1* (leading to the quadruple mutant) increased the sensitivity of the triple mutant. This observation is further confirmed by the ectopic integration of *CEK1* in the *cek1 cek2 mkc1 hog1* strain, which restored the phenotype. This result is in line with the previous hypothesis that the *hog1* resistant phenotype to cell wall damage was mainly due to Cek1 hyperactivation [38]. Nevertheless, other elements or mechanisms besides Cek1 hyperactivation may be implicated in the resistance of *hog1* cells to Congo red as the deletion of *HOG1* alleviates the hypersensitivity caused by the absence of a functional Cek1/Cek2 and Mkc1 pathways. As shown in Figure 4B, the triple mutant *cek1 cek2 mkc1*, similar to our previous observation with *cek1 mkc1* and *cek2 mkc1* strains [37], is extremely sensitive to Congo red, while *cek1 cek2 mkc1 hog1* cells were not as sensitive and were even able to proliferate after prolonged incubation times at higher concentrations of stressing agents. As expected, the ectopic expression of *HOG1* increased the sensitivity of the quadruple mutant to the levels of its corresponding strain *cek1 cek2 mkc1*.

In a preliminary assay with nikkomycin Z we were also able to assess whether the previously described resistant phenotype of *hog1* strain was Cek1 dependent. This compound also targets the fungal cell wall through inhibition of chitin synthesis, acting as a competitive analogue of chitin synthase substrate UDP-N-acetylglucosamine [49]. All strains lacking the Hog1 kinase were resistant to nikkomycin Z (Appendix A) suggesting that no other MAPK is involved in counterbalancing the phenotype. Although the quadruple mutant behaved similarly to the wild type, the ectopic expression of *HOG1* remarkably induced strain sensitivity to nikkomycin Z and an increase in resistance was observed upon integration of the other kinases. Interestingly, the deletion of *CEK1* and/or *CEK2* did not significantly alter the behavior of *C. albicans* in the presence of nikkomycin Z, despite their clear role in cell wall biogenesis and in the response to cell wall perturbing agents. In fact, the response to this compound seems to highly depend on Mkc1 and Hog1. According to our results, the *mkc1* phenotype could be caused by a compensatory mechanism involving the HOG pathway as all *mkc1* mutants showed a sensitive phenotype to this drug that was only reverted by the additional deletion of *HOG1*. This behavior was also observed in response to caspofungin [48].

### 3.3. Hog1 Is the Key MAPK for C. albicans Morphological Transitions

MAPK pathways are involved in *C. albicans* morphogenesis and, particularly, in the yeast-to-hypha transition. Hog1 is a well-characterized repressor of filamentation and essential for the development of chlamydospores [15,25], while Mkc1 is implicated in biofilm formation and in the morphologic transition to hyphae on solid medium [28,50]. Cek1 has an important role on *C. albicans* vegetative growth and in the dimorphic yeast-to-hypha transition in response to several stimuli [41].

We were interested in analyzing our MAPK-defective strains regarding the morphogenetic program as well as determining the complementary roles or unveiling synergies between the MAPK signaling pathways. For this purpose, we analyzed the morphology of fungal cells grown in 5% serum (under weak filament inducing conditions) or in 100% serum (strong inducing conditions) at 30 °C in liquid media by microscopic analysis.

As shown in Figure 5, the quadruple mutant (*cek1 cek2 mkc1 hog1*) is able to filament after 5 h of growth in 5% serum and even on YPD medium, therefore resembling the *hog1* phenotype. The dominant repressive role of Hog1 on morphogenesis is demonstrated by the hyphal morphology presented by all *hog1* mutants under non-inducing conditions (Figure 5). The deletion of *CEK1* in the *hog1* background (*cek1 hog1* and *cek1*
*cek2 hog1*) seems to slightly inhibit filamentation. The same is true when comparing the *hog1 mkc1* with *cek1 mkc1 hog1* strain [48]. This was somewhat expected due to the complementary roles these two MAPKs exert on morphological transitions. All mutants analyzed were able to form hyphae as observed upon 24 h of growth in 100% serum (Appendix A).

We also analyzed the ability of our multiple deletion strains to generate chlamydospores (Figure 6).

Neither Cek1, Cek2 nor Mkc1 is necessary for the formation of these structures. In fact, *cek2* and *cek1 cek2* seem to present clusters of cells that could derive from an increased number of chlamydospores. In clear contrast, deletion of *HOG1* completely abolished this morphotype, confirming previous studies that indicate the essential role of *HOG1* in this process. None of the mutants in a *hog1* background (including *cek1 hog1, cek2 hog1* and *cek1 cek2 hog1*) were able to form chlamydospores, even after a prolonged incubation period and maintenance under inducing conditions (microaerophilia, low temperature and darkness). This was unexpected as the double mutant *hst7 hog1*, defective in Cek1/Cek2 activation, showed chlamydospore formation [38]. The quadruple mutant was unable to develop chlamydospores.

### 3.4. The Absence of MAPKs Triggers White-Opaque Switching in Rich Medium in a Hog1-Dependent Manner

During our phenotyping analysis we observed that three-day-old colonies from the quadruple mutant, grown in YPD solid medium at room temperature (<21 °C), presented grey sectors emerging from the white standard colonies, which potentially indicates a phenotypic switch from white-to-opaque cells. In order to confirm the presence of opaque cells amongst our colonies, we prepared cell suspensions of two independent strains defective in all four MAPKs (*cek1 cek2 hog1 mkc1* and *cek1 cek2 mkc1 hog1*) obtained from white colonies previously grown on YPD solid medium at 30 °C (with no grey sectors). These cell suspensions were inoculated on YPD phloxine B plates (which are used to discriminate between white and opaque sectors), and pink sectors were observed after 7 days’ growth at room temperature (Figure 7A), supporting the idea of a spontaneous switching. The colonial morphology of these cells resembled the opaque phenotype: flattened colonies with a matte appearance, contrasting to convex and shiny white colonies.

As opaque cells are highly unstable at physiological temperatures [51,52], mass switching of opaque cells to white cells can be induced by a temperature shift from (below) 25 °C to 37 °C [53]. We thus analyzed the colonies phenotype at different temperatures. Opaque cells from distinguishable dark-pink colonies of the YPD plates grown at RT were re-inoculated on fresh YPD (with phloxine B) and incubated at 37 °C, 30 °C or below 21 °C (RT). The opaque state of the quadruple mutant was stable at RT as almost all colonies maintained the dark-pink coloration characteristic of this phenotype. However, previously opaque cells turned into the white phenotype at 37 °C and no opaque colonies, nor opaque sectors, were seen even after a longer incubation period. Cells also switched back from opaque to white gradually when incubated at 30 °C and almost every colony presented a mixture of white and opaque cells seen as perfectly differentiated colony sectors (Appendix A). White-opaque switching of the quadruple mutant was unexpected as this transition is under the control of the mating type a1/α2 repressor; we thus suspected that during the generation of the quadruple strain, a conversion to the homozygous type had occurred, probably due to the stress imposed on cells as it can promote genetic rearrangements [54]. Through PCR amplification of the MTL locus, we were able to confirm that this strain became in fact MTL a/a.

This, in turn, allowed us to better understand the contribution of each MAPK to the observed phenotype. We integrated each of the four MAPKs into the *ADH1* locus of the quadruple homozygous mutant and analyzed the obtained strains for their switching ability on YPD solid medium at RT (Figure 7B). The ectopic expression of *CEK1* or *MKC1* did not affect switching and the presented phenotype was similar to that of the quadruple mutant (100% of the analyzed colonies with opaque sectors). However, as evidenced in Figure 7C, the overexpression of *CEK2* led to a decrease in the number of cells with opaque sectors (from 100% of the quadruple mutant to 67%) and *HOG1* ectopic expression completely abolished white-opaque switching from the quadruple mutant (100% white colonies). These results suggest that Hog1 and Cek2 (but not Cek1 nor Mkc1) are involved in the white-opaque transition of *C. albicans*.

## 4. Discussion

In this work we have generated and characterized strains defective in three or in all four major MAPKs of *C. albicans*, to further advance our understanding about the complex crosstalk among these pathways [33,34,35,55]. These mutants allowed us to answer two main questions: Can *C. albicans* survive without a functional MAPK network? Does *C. albicans* depend on a specific MAPK to respond and adapt to stress conditions and, if so, to what extent? We show here that *cek1 cek2 mkc1 hog1* cells are viable and show no evident growth defects under standard laboratory *in vitro* conditions, although they are highly sensitive to stress.

We were able to confirm the major role of Hog1 in *C. albicans* during oxidative and osmotic stress; cells with a functional HOG pathway are able to efficiently grow under these conditions and the ectopic expression of *HOG1* in the quadruple mutant completely overcomes the sensitivity of this mutant. However, while the quadruple mutant presents the same phenotype as a *hog1* single mutant in the presence of oxidants, it displays an increased sensitivity to osmotic stress similar to what is observed in cells which are also defective in *CEK1* [37,44]. The survival and proliferation of *C. albicans* under oxidative conditions on solid media is mainly dependent on Hog1, even though other kinases show altered levels of phosphorylation upon oxidative challenge [29,36,38]. Interestingly, we cannot discard the involvement of other MAPKs in the susceptibility to macrophage killing as the quadruple mutant has an increased susceptibility compared to triple and double mutations, even though *hog1* is the only single mutant whose killing is increased compared to wild-type cells [43]. This result is in line with both the existence of multiple mechanisms involved in killing by macrophages and cell wall changes (as revealed by the accessibility to specific epitopes [31]) in these strains that may influence interactions with immune cells [56].

Oxidative stress has also been associated with the mechanism of action of amphotericin B and fluconazole [45,46,57]. Our results are in agreement with this, as strains defective in *HOG1* (which are highly susceptible to oxidative stress), are also sensitive to these antifungals. It has recently been shown that treatment with amphotericin B results in the phosphorylation of Hog1 and that the absence of this stress kinase increases fungal susceptibility to this antifungal compound [58]. Strikingly, *cek1*, *cek2* and *cek1 cek2* double mutants were consistently less susceptible to amphotericin B. Cell wall alterations caused by the deletion of these kinases [34,37,59] could alter the accessibility of the antifungal to the membrane. While we cannot discard this, we are more inclined to the hypothesis of a more permeable, loose, and weakened cell wall in *cek1/2* mutants.

The *C. albicans hog1* strain is more resistant than the wild type to compounds that affect the fungal cell wall, such as Congo red, nikkomycin z and to a lesser extent to calcofluor white [25]. This is consistent with the role of the HOG pathway in controlling (at least partially) chitin synthesis [60] and to the sustained activation of the Cek1 kinase observed in *hog1* mutants [29,61]. In fact, the deletion of *HST7* (Cek1 MAPK kinase) or *CEK1* itself (but not *CEK2*) in a *hog1* background suppressed the resistant phenotype [37,38]. However, when analyzing strains defective in three or four MAPKs, we were able to reveal the putative existence of a Cek1-independent mechanism to explain *hog1* resistance to cell wall interfering compounds. First, deletion of *HOG1* alleviates the Congo red hypersensitivity caused by the absence of a functional Cek1/Cek2 and Mkc1 pathways and the ectopic expression of Hog1 in the quadruple mutant increases its sensitivity. Second, the deletion of *CEK1* in strains with a *hog1* background was not sufficient to suppress their resistant phenotype to nikkomycin Z. Therefore, other elements or mechanisms, besides Cek1 hyperactivation, may be implicated in resistance of the *hog1* mutant to cell wall perturbing agents. The same is true for chlamydospore formation as none of the strains defective in Hog1 kinase were able to form chlamydospores. It has been suggested that Hog1 would have an inducing role on this phenotype through the hyperactivation of Cek1 due to the wild-type-like phenotype displayed by the *hog1 hst7* strain [38]; however, this hypothesis could not be tested at the time due to the lack of a *cek1 hog1* double mutant. Our results support a critical role for Hog1—discarding any evident role for Cek1, Cek2 and Mkc1—in the formation of chlamydospores.

Our results also confirm a major role for *HOG1* in repressing hyphal formation as all mutants with a *HOG1* deletion were shown to filament in non-inducing (or mildly-inducing-) conditions. Such effects have been described to be dependent on the TOR pathway, which connects nutrient sensing and MAPK signaling. Under low nutrient availability, specific tyrosine phosphatases dephosphorylate and thus deactivate Hog1, which in turn suppresses the potential of Hog1 to repress hyphal elongation; this process occurs via dissociation of the Sko1 complex from the promoter of Brg1, a positive regulator of filamentation [62]. Coherently, strains defective in elements from the HOG pathway express Brg1 independently of rapamycin. Our data also indicate that none of the MAPKs, either alone or in combination, are essential for the dimorphic transition under serum, inducing conditions which may be relevant to disseminated infection. This is consistent with the involvement of several pathways in the dimorphic transition [63] and the apparent redundancy between them.

*cek1 cek2 mkc1 hog1* mutants were able to undergo white-opaque switching with high frequency on YPD medium. This transition occurred at low temperature (<21 °C) and without any other specific environmental requirement. The involvement of MAPK signaling in this process makes sense in the context of their role as sensor mechanisms of the environment. In fact, environmental conditions seem to substantially regulate white-opaque switching. Its efficiency and the stability of the opaque phenotype is highly affected by temperature [51,52,64], by the level of CO_2_ [65,66], by the sugar source [67] and also by genotoxic and oxidative stress [68], UV radiation, in vitro oxidants and white blood cell metabolites [69]. Some of these stimuli are known to trigger activation of MAPKs and it is therefore conceivable that evolution has coupled MAPK cascades to enable a precise timing of white-opaque switching. In fact, recent reports have shown that *hog1* mutants in a/a background (homozygous for MTL), as well as other strains defective in upstream elements of the HOG pathway, display an increased switching phenotype that is dependent on Wor1 [70,71], the master regulator of white-opaque switching [72,73]. We were able to confirm this observation as the ectopic expression of Hog1 in *cek1 cek2 mkc1 hog1* cells completely abolished the opaque state of this strain. We were also able to implicate, at least partially, Cek2 (but not Cek1) in this process. It is surprising and unexpected that the expression of *CEK1* in a quadruple mutant did not block the formation of opaque sectors. Ramírez-Zavala and co-workers have assessed the switching frequency of a mating competent strain (WO-1–MTLα/α) overexpressing a hyperactive allele of Ste11 [74], the MAPK kinase kinase of the Cek1/Cek2 mediated pathway. The white-opaque switching is increased in this strain, by an unknown physiological signal that is transmitted along the pathway and it was shown to require the downstream MAPK Cek1 (but not Cek2) and its target transcription factor Cph1. Even though certain a/α strains (heterozygous for MTL) are capable of white-opaque switching [75], homozygosis still seems to be an important requirement for this process. The epigenetic switch is controlled by Wor1 (amongst other transcription factors) whose expression is repressed by MTL a1/α2 repressor and thus the phenotypic transition, as well as mating, is inhibited in cells heterozygous for the MTL locus. This is consistent with the fact that the minority of natural isolates capable of undergoing the white-opaque switch in vitro were found to be a/a or α/α [19,73,76]. Our results confirm that *HOG1* is the main kinase involved in white-opaque switching but also reveal a role for Cek2 (but not Cek1) in this process. Functional differences between Cek1 and Cek2 have been detected by our group [37], and recently it has been shown that the phosphorylation and nuclear localization in response to pheromone is different between both kinases [77].

Our newly described *cek1 cek2 mkc1 hog1* strain is an interesting and powerful tool in different areas of research. As a strain completely devoid of MAPKs could be used to perform synthetic biology studies with mammalian or other fungal MAPKs. Given its likely lack of virulence in animal hosts (all *hog1*, *cek1* and *mkc1* show reduced virulence in the mouse systemic animal model) [15,27,31,41]), it could potentially be used as an attenuated live vaccine. Finally, we envision that it will be also an attractive tool for the analysis of genetic recombination in this fungus induced by stress [22], a phenomenon that seems to be of great clinical significance [78].

## Figures and Tables

**Figure 1 jof-06-00230-f001:**
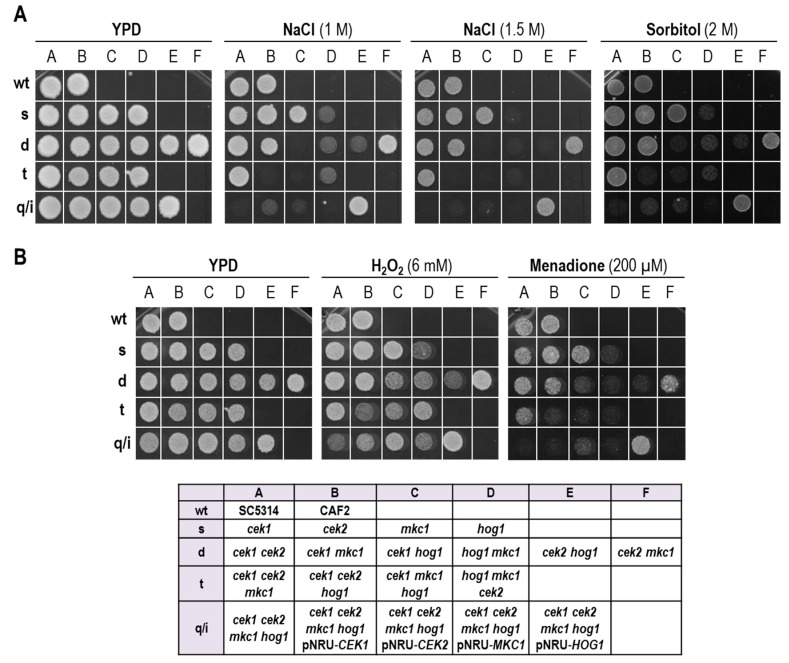
Comparative growth of mitogen-activated protein kinase (MAPK) mutants in the presence of osmotic and oxidative stress. 10^4^ cells from a set of MAPK mutants grown overnight at 37 °C in yeast extract–peptone–dextrose (YPD) were spotted on solid agar plates of YPD medium supplemented or not (YPD) with NaCl, sorbitol, H_2_O_2_ or menadione at the indicated concentrations. Plates were incubated at 37 °C and pictures were taken at 24 h (**B**) or 48 h (**A**). A diagram of the strains’ locations within the plate is shown. wt: wild type; s: single mutants; d: double mutants; t: triple mutants; q: quadruple mutant; i: quadruple mutants with an ectopically integrated MAPK.

**Figure 2 jof-06-00230-f002:**
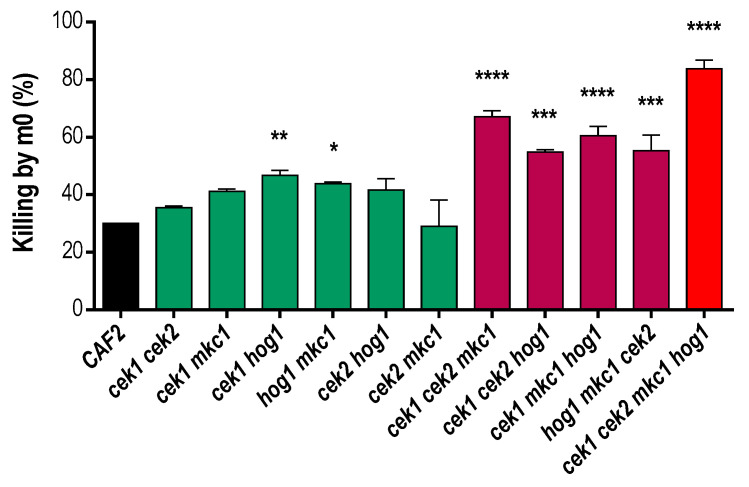
Susceptibility of MAPK mutants to macrophage-mediated killing. Yeast cells were added to a confluent monolayer of murine macrophages from cell line RAW 264.7 at MOI 1:20 (yeast:m0) and co-incubated for 4 h at 37 °C and 5% CO_2_. Killing percentages were assessed after CFU counting of fungal cells obtained upon interaction and related to fungal cells assayed without macrophages (positive control–0% killing). All strains were normalized to the wild type (CAF2) with a given value of 30% killing. Error bars represent the standard deviation of duplicates. * *p* < 0.05; ** *p* < 0.01; *** *p* < 0.001; **** *p* < 0.0001 (one-way ANOVA, multiple comparison tests related to CAF2).

**Figure 3 jof-06-00230-f003:**
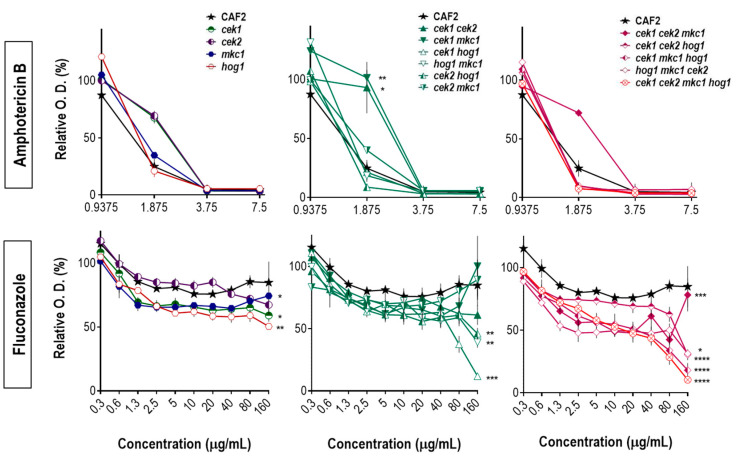
Sensitivity of MAPK mutants to antifungal treatment. Cells from cultures grown overnight at 37 °C were inoculated in a 96-well plate at final O.D. = 0.025 (10^5^ cells). Amphotericin B or fluconazole was added to the plates in increasing 2-fold concentrations ranging from 0.030 µg/mL to 15 µg/mL (shown only 0.94 µg/mL to 7.5 µg/mL-amphotericin B) or 0.32 µg/mL to 160 µg/mL (fluconazole). Inoculated plates were incubated at 37 °C for 24 h prior to O.D. assessment. Each strain has been normalized to its positive control (without antifungal) and the growth inhibition curve is shown. Error bars represent the standard error of the mean from two independent experiments. * *p* < 0.05; ** *p* < 0.01; *** *p* < 0.001; **** *p* < 0.0001 (two-way ANOVA multiple comparison tests related to CAF2).

**Figure 4 jof-06-00230-f004:**
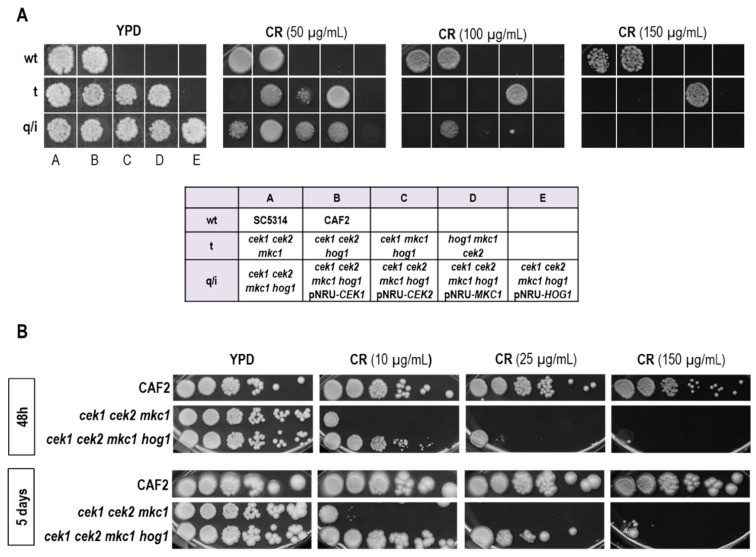
Comparative growth of MAPK mutants in the presence of Congo red. (**A**) Cells from a set of MAPK mutants from overnight grown cultures were spotted on solid agar plates of YPD medium supplemented or not (YPD-control) with different Congo red concentrations. Plates were incubated for 48 h at 37 °C. (**B**) 10^5^ cells and tenfold dilutions from overnight grown cultures were spotted on solid agar plates of YPD medium supplemented or not (control) with increasing concentrations of Congo red. Plates were incubated at 37 °C for 48 h or five days. wt: wild type; t: triple mutants; q: quadruple mutant; i: quadruple mutants with an ectopically integrated MAPK.

**Figure 5 jof-06-00230-f005:**
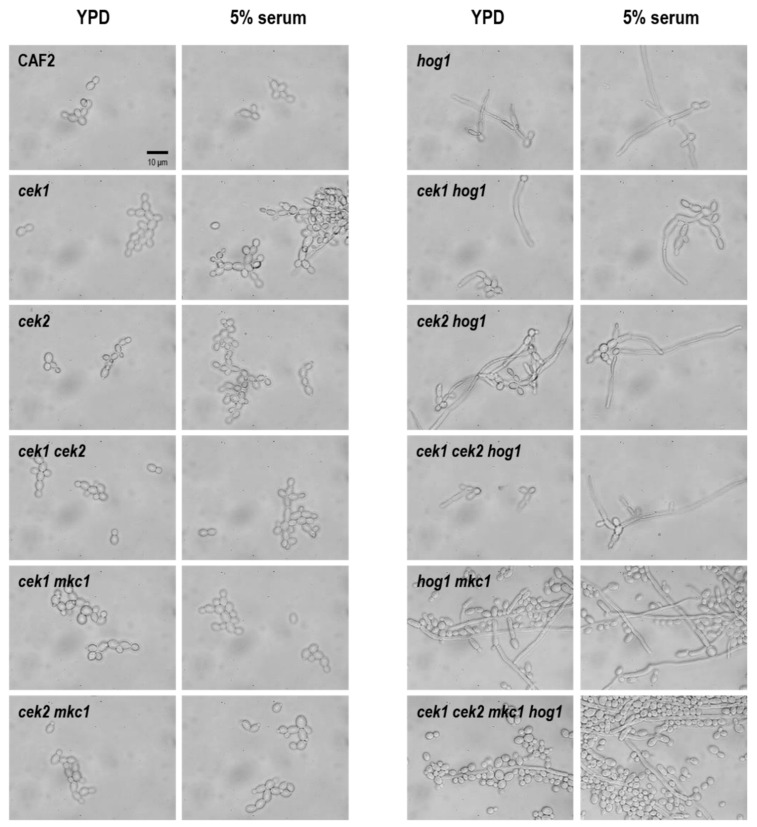
Yeast-to-hypha transition in liquid medium. 10^6^ cells from overnight cultures were inoculated in YPD medium or 5% serum. Samples were observed after 5 h of growth at 30 °C and representative pictures are shown. All pictures were taken at the same magnification. Scale bar = 10 µm.

**Figure 6 jof-06-00230-f006:**
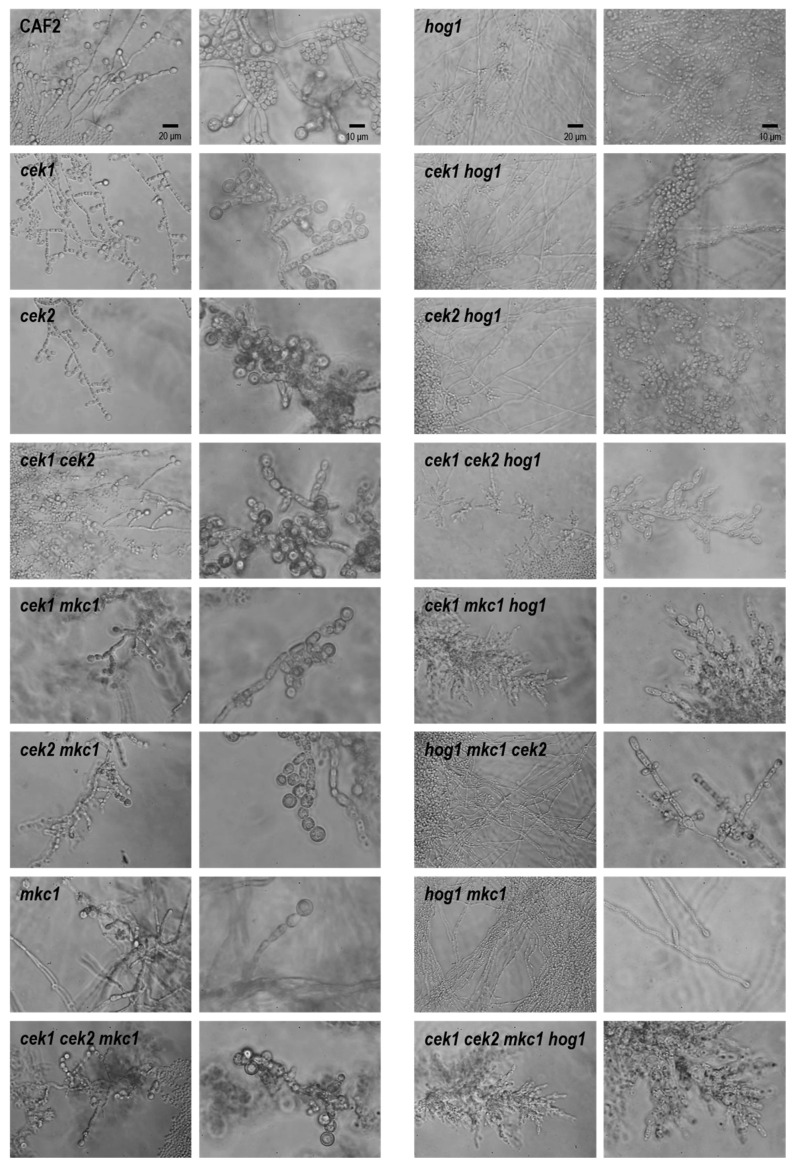
Twenty-five to fifty cells from stationary grown cultures were inoculated on Corn Meal Agar medium supplemented with Tween 80. Chlamydospores were observed between 4 and 7 days upon incubation at 24 °C in dark and microaerophilic conditions. Representative pictures are shown. Pictures in the same column were taken at the same magnification. Scale bar is shown in the first row.

**Figure 7 jof-06-00230-f007:**
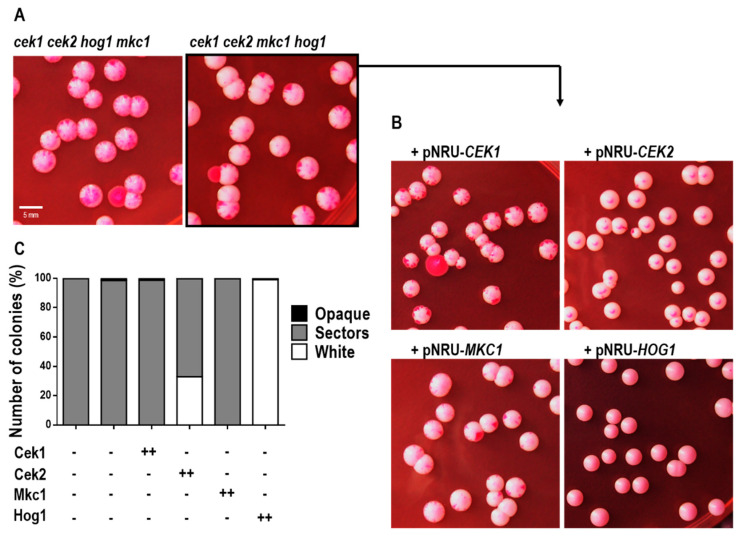
White-opaque switching of *C. albicans* strains lacking MAPKs. The quadruple mutant (*cek1 cek2 mkc1 hog1*) was transformed with the plasmid pNRU carrying the *CEK1*, *CEK2*, *MKC1* or *HOG1* genes and the obtained strains were analyzed. Cells from white colonies grown on YPD at 30 °C were inoculated on YPD agar plates supplemented with phloxine B. (**A**) and (**B**) Representative pictures of colonies morphology of the indicated strains, from four independent experiments, taken after 8 days growth at room temperature (RT) (<21 °C). (**C**) Summary graphic showing the percentage of white, opaque or white colonies with opaque sectors observed upon 12 days of growth at RT. At least 400 colonies from two independent experiments were analyzed. - stands for gene disruption and ++ for strains expressing ectopically (at *ADH1* locus) a single copy of the MAPK gene. All pictures were taken at the same magnification. Scale bar = 5 mm.

**Table 1 jof-06-00230-t001:** Strains of *C. albicans* used in this work.

Strain Name	Background Strain and Genotype	Source
SC5314 (wt)	Clinical isolate	[40]
CAF2 (wt)	*ura3* *Δ::imm434/ura3* *Δ::imm434-URA3*	[39]
CAI4	*ura3* *Δ::imm434/ura3* *Δ::imm434*	[39]
*cek1*	[CAI4] *cek1**Δ::hisG-URA3-hisG/cek1**Δ::hisG*	[41]
*cek2*	[CAI4] *cek2::cat-URA3-cat/cek2::cat*	[37]
*mkc1*	[CAI4] *mkc1::hisG-URA3-hisG**/**mkc1::hisG*	[27]
*hog1*	[CAI4] *hog1::hisG-URA3-hisG/hog1::hisG*	[42]
*cek1 cek2*	[CAI4] *cek1::hisG**/**cek1::hisG cek2::cat-URA3-cat**/**cek2::cat*	[37]
*cek1 mkc1*	[CAI4] *cek1::hisG**/**cek1::hisG mkc1::hisG-URA3-hisG**/**mkc1::hisG*	[37]
*cek1 hog1*	[CAI4] *cek1::hisG**/**cek1::hisG hog1::hisG**/**hog1::hisG ARD1**/**ard1::FRT SAP2pr-FLPURA3*	[43]
*hog1 mkc1*	[CAI4] *hog1::hisG/hog1::hisG mkc1::hisG-URA3-hisG/mkc1::hisG*	[37]
*cek2 hog1*	[CAI4] *cek2::cat/cek2::cat hog1::hisG-URA3-hisG/hog1::hisG*	[37]
*cek2 mkc1*	[CAI4] *cek2::cat/cek2::cat mkc1::hisG-URA3-hisG/mkc1::hisG*	[37]
*cek1 cek2 mkc1*	[CAI4] *cek1::hisG**/**cek1::hisG cek2::cat**/**cek2::cat* *mkc1::hisG-URA3-hisG**/**mkc1::hisG*	This work
*cek1 cek2 hog1*	[CAI4] *cek1::hisG**/**cek1::hisG cek2::cat**/**cek2::cat* *hog1::hisG-URA3-hisG/hog1::hisG*	This work
*cek1 mkc1 hog1*	[CAI4] *cek1::hisG**/**cek1::hisG mkc1::hisG**/**mkc1::hisG* *hog1::hisG-URA3-hisG/hog1::hisG*	This work
*hog1 mkc1 cek2*	[CAI4] *hog1::hisG/hog1::hisG mkc1::hisG/mkc1::hisG* *cek2::cat-URA3-cat/cek2::cat*	This work
*cek1 cek2 hog1 mkc1*	[CAI4] *cek1::hisG**/**cek1::hisG cek2::cat**/**cek2::cat* *hog1::hisG/hog1::hisG* *mkc1::hisG-URA3-hisG**/**mkc1::hisG*	This work
*cek1 cek2 mkc1 hog1*	[CAI4] *cek1::hisG**/**cek1::hisG cek2::cat**/**cek2::cat* *mkc1::hisG**/**mkc1::hisG* *hog1::hisG-URA3-hisG/hog1::hisG*	This work
*cek1 cek2 mkc1 hog1* pNRU-*CEK1*	[CAI4] *cek1::hisG**/**cek1::hisG cek2::cat**/**cek2::cat* *mkc1::hisG**/**mkc1::hisG* *hog1::hisG/hog1::hisG* *ADH1/adh1::tTA pTet-CEK1-myc-URA3*	This work
*cek1 cek2 mkc1 hog1* pNRU-*CEK2*	[CAI4] *cek1::hisG**/**cek1::hisG cek2::cat**/**cek2::cat* *mkc1::hisG**/**mkc1::hisG* *hog1::hisG/hog1::hisG* *ADH1/adh1::tTA pTet-CEK2-myc-URA3*	This work
*cek1 cek2 mkc1 hog1* pNRU-*MKC1*	[CAI4] *cek1::hisG**/**cek1::hisG cek2::cat**/**cek2::cat* *mkc1::hisG**/**mkc1::hisG* *hog1::hisG/hog1::hisG* *ADH1/adh1::tTA pTet-MKC1-myc-URA3*	This work
*cek1 cek2 mkc1 hog1* pNRU-*HOG1*	[CAI4] *cek1::hisG**/**cek1::hisG cek2::cat**/**cek2::cat* *mkc1::hisG**/**mkc1::hisG* *hog1::hisG/hog1::hisG* *ADH1/adh1::tTA pTet-HOG1-myc-URA3*	This work

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
