# Peer review of "Characterization of a Candida albicans Mutant Defective in All MAPKs Highlights the Major Role of Hog1 in the MAPK Signaling Network"

_jof, 2020, doi:10.3390/jof6040230_

Round 1

Reviewer 1 Report

Reference#: Jof-950585 peer-review- v1

Characterization of a Candida albicans mutant defective in all MAPKs highlights the major role of  Hog1 in the MAPK signaling network. Minor comments for this manuscript:

  1. Page 3 line 91: YPD medium,appears first time, please write full name.
  2. Page 3 line 93: “4ºC”, please corrected.
  3. Fig (B): Doesn't seem to match the expected results? Please explain.

Author Response

  1. "YPD medium" replaced for "Yeast extract Peptone Dextrose (YPD) medium"
  2. corrected
  3. Number of figure is missing.

Reviewer 2 Report

The authors demonstrate that disruption of multiple mitogen-activated protein kinases in Candida albicans is not essential for growth. However this quadruple mutant which lacks the four MAPKs in C. albicans resulted in hypersensitivity to osmotic and oxidative stress. Antifungal susceptibility testing determined that it was deletion of Hog1 in the quadruple mutant that resulted in hypersensitivity to amphotericin B and fluconazole. Exposure to cell wall stress determined Hog1 resistance to Congo red which was independent of Cek1. The quadruple mutant was also able to filament under non-inducing conditions but is unable to form chlamydospores. In addition the authors show that deletion of all four MAPKs leads to a high frequency of white opaque switching which is dependent on Hog1.

Points to address:

It would be useful to include the westerns, southerns or PCRs used to confirm the construction of the new strains made in this study in the supplementary information.

In Figure 2 & 4 why were the single and double mutants not included? I thought it would have been beneficial to see these mutants side by side with the others to directly compare them all, just like they were presented in some of the other figures.

The authors should be cautious when referring to strains lacking CEK1 and/or CEK2 as more resistant to amphotericin B (line 195). Reduced susceptibility would be a better term as there is only 1-2 doubling dilutions difference between these strains and the wild-type.

Why was caspofungin not tested as an antifungal that targets the cell wall and tested in the say way as Amphotericin B and Fluconazole?

Scale bars should be added to Figure 5, 6 & 7

Why was 30oC used rather than 37oC for the filament inducing conditions?

Figure 6 – The panels for the different time points should be labelled as 4 and 7 days. Would it be possible to replace the images for the hog1, cek1hog1, cek2hog1 and cek1cek2mkc1hog1 for better images because I found it difficult to see the cells clearly in them.  

The complementary data for Figure 5 of the cells grown in 100% serum could be added to the supplementary data for reference.

The legend of Figure 7 is missing a description of part A.

Author Response

1. "It would be useful to include the westerns, southerns or PCRs used to confirm the construction of the new strains made in this study in the supplementary information."

We have included, in the supplementary information, a figure with the Southern blot analysis (using four probes, one for each MAPK) of the quadruple mutant lineage: cek1 – cek1cek2 – cek1cek2mkc1 – cek1cek2mkc1hog1.

2. "In Figure 2 & 4 why were the single and double mutants not included? I thought it would have been beneficial to see these mutants side by side with the others to directly compare them all, just like they were presented in some of the other figures."

We have not included those data as it has been previously published by our group: "Differential susceptibility of mitogen-activated protein kinase pathway mutants to oxidative-mediated killing by phagocytes in the fungal pathogen Candida albicans" (Arana, et al. 2007) and "Complementary roles of the Cek1 and Cek2 MAP kinases in Candida albicans cell-wall biogenesis" (Correia2016).

3. "The authors should be cautious when referring to strains lacking CEK1 and/or CEK2 as more resistant to amphotericin B (line 195). Reduced susceptibility would be a better term as there is only 1-2 doubling dilutions difference between these strains and the wild-type."

The reviewer is right, changes have been made.

4. "Why was caspofungin not tested as an antifungal that targets the cell wall and tested in the say way as Amphotericin B and Fluconazole?"

We have tested caspofungin the same way as nikkomycin Z (line 245). The results obtained were similar for both antifungals regarding mkc1 sensitivity and its rescue by the additional deletion of HOG1. As it was a preliminary assay and it didn´t add any novel information nor it sustained the fact that hog1 resistance is Cek1-independent (since hog1 behaves similarly to the wild-type in the presence of caspofungin) we decided not to include this figure in the manuscript. It can however be added if the reviewer believes it would be a better option.

5. "Scale bars should be added to Figure 5, 6 & 7"

Added.

6. "Why was 30oC used rather than 37oC for the filament inducing conditions?"

Our group has shown that Hog1 is a repressor of filamentation as hog1 mutants are able to filament under weak inducing conditions such as 1% serum at 37ºC (Alonso-Monge, et al. 1999). Therefore, we have used 30ºC instead of 37ºC to reveal putative hyperfilamentous phenotype of the selected mutants and to highlight small differences amongst mutants with an hog1 background.

7. "Figure 6 – The panels for the different time points should be labelled as 4 and 7 days. Would it be possible to replace the images for the hog1, cek1hog1, cek2hog1 and cek1cek2mkc1hog1 for better images because I found it difficult to see the cells clearly in them."

The panels do not necessarily correspond to different time points. Images were taken between day 4 up to 7. As we were mainly interested in knowing whether these mutants were able or not to form chlamydospores, we didn´t explore morphological or timing details. Unfortunately, we don´t have better images to replace the ones in the figure. However, we believe it is clear that the referred strains do not form chlamydospores.

8. "The complementary data for Figure 5 of the cells grown in 100% serum could be added to the supplementary data for reference."

Added to the supplementary data - Figure S3: Yeast-to-hypha transition in 100% serum.

9. "The legend of Figure 7 is missing a description of part A."

A) and B) refer to the same experiment with different strains. Legend was altered to become more clear.